# Serological Response to SARS-CoV-2 after COVID-19 Vaccination in Lung Cancer Patients: Short Review

**DOI:** 10.3390/vaccines11050969

**Published:** 2023-05-11

**Authors:** Ananda M. Rodilla, Sooyun Tavolacci, Jazz Cagan, Tanay Shah, Sandeep Mittan, Philip C. Mack, Fred R. Hirsch

**Affiliations:** 1Center for Thoracic Oncology, Tisch Cancer Institute, Icahn School of Medicine at Mount Sinai, New York, NY 10029, USA; 2Department of Obstetrics, Gynecology and Medical Oncology, Montefiore Medical Center, The University Hospital for Albert Einstein College of Medicine, New York, NY 10461, USA

**Keywords:** lung cancer, SARS-CoV-2 vaccine, immunogenicity

## Abstract

In comparison to the general population, lung cancer patients are more likely to suffer from severe Coronavirus disease (COVID-19) and associated mortality. Considering this increased risk, and in order to prevent symptoms and severe disease, patients with lung cancer have been prioritized for COVID-19 vaccination primary and booster doses. Despite this, the pivotal clinical trials did not include these patients, which leaves open questions regarding vaccine efficacy and humoral immune response. This review outlines the findings of recent investigations into the humoral responses of lung cancer patients to COVID-19 vaccination, particularly the primary doses and first boost.

## 1. Introduction

The severe acute respiratory syndrome coronavirus 2 (SARS-CoV-2) was first identified in China in December 2019. It quickly spread to other nations, leading the World Health Organization (WHO) to declare COVID-19 a global pandemic in March 2020. Compared to the general population, patients with cancer are more likely to suffer from severe Coronavirus disease (COVID-19) and associated mortality [1]. The active malignancy, coupled in many instances with advanced age and/or other comorbidities, is considered a risk factor for severe COVID-19 development [2]. In addition, in the early months of the pandemic, there was a substantial decline in cancer screening, which delayed ongoing or planned therapy and increased the risk of poor outcomes [3]. Earlier reports identified patients with lung cancer (LC) as a vulnerable population to COVID-19 due to their high rates of SARS-CoV-2 infection, more severe course of COVID-19 disease, and predicted fatality rate of more than 30%, especially during the first wave [4,5,6,7].

In light of this increased risk, preventing the onset of severe COVID-19 is essential in these patients, and the most efficient way to achieve this goal is through vaccination. Fortunately, the SARS-CoV-2 vaccinations were made widely available after fast development in December 2020. Over the past two years, dozens of coronavirus vaccines have entered clinical trials. In the U.S., the most widely used vaccines against COVID-19 are the mRNA vaccines developed by Pfizer-BioNTech and Moderna (BNT162b2 and mRNA-1273, respectively). Worldwide, an array of adenovirus-vectored vaccines was implemented, including the Oxford University–AstraZeneca (ChAdOx1 nCoV19), the Janssen Ad 26 (Ad26.COV2-S), and the Sputnik V vaccines (Gam-COVID-Vac) [8]. In the randomized Phase III clinical trials, these vaccines proved to be safe and effective in the great majority of the participants (>90%) [9,10,11,12,13]. However, within months of vaccination, efficacy gradually attenuated [14,15,16,17], and newly emerging variants of concern (VOC) at the time raised concerns regarding the escape from pre-existing immunity to SARS-CoV-2 [18,19]. To address these concerns, the United States FDA authorized booster shots for certain high-risk groups in late September 2021, about 8 months after the initial vaccination [20,21,22]. In several countries, additional boosters were recommended for elderly and immunocompromised individuals at risk of severe disease [23,24], being the case for cancer patients in the United States. Moreover, after the emergence of Omicron sub-variants, the FDA authorized bivalent mRNA vaccines for COVID-19 in the fall of 2022 to boost protection against these sub-lineages and enhance defense against severe disease [25].

Once the SARS-CoV-2 vaccine became available, cancer patients were prioritized for both primary and booster vaccinations against COVID-19 [26,27]. These patients, however, were not a focus of the pivotal clinical trials, leaving open questions regarding vaccine effectiveness, humoral immune responses, and potential side effects related to vaccines in this vulnerable population. Consequently, these questions have been inferred from small prospective observational studies examining immune responses in various cancer populations.

The purpose of this review is to provide an overview of the findings of studies in patients with LC on the humoral immunogenicity of the SARS-CoV-2 vaccine primary and first booster doses.

## 2. SARS-CoV-2 Vaccines Immunogenicity in Patients with LC or Thoracic Cancer

The majority of research on SARS-CoV-2 vaccination in LC patients has evaluated the capacity of vaccines to elicit immunological responses via the measurement of SARS-CoV-2 spike-binding antibodies, expressed as seroconversion rates or mean antibody titers at various time points after vaccination. To find available studies assessing serological response in LC patients, Embase and PubMed were searched with the combination of the following terms: SARS-CoV-2; COVID-19; immunogenicity; antibodies; and lung cancer. Table 1 summarizes the key information from nine eligible studies that only included LC patients. This information includes vaccination types, test groups, histological classification, age, and treatment stratification. Additionally, Table 1 presents the endpoints of the studies, the assay used, and the main findings related to the humoral response. The nine studies examined a variety of COVID-19 vaccines, including mRNA vaccines such as BNT162b2 (Pfizer-BioNTech) and mRNA-1273 (Moderna), as well as viral vector-based vaccines such as AstraZeneca’s AZD1222 and Johnson & Johnson’s Ad26.COV2.S. The proportion of each vaccine used differed among the studies, but BNT162b2 and mRNA-1273 were the most commonly investigated. Each study used different assays and units to report serological response, including seroconversion rates or median antibody titers in units per milliliter (U/mL), arbitrary units (AU/mL), international units per milliliter (IU/mL), or standardized binding antibody units (BAU/mL). The time interval from vaccination to serological response assessment was also diverse among the studies, with most studies reporting one month after vaccination, while others ranged from 2 weeks to 3–6 months. All these disparities make it challenging to compare and synthesize quantitative results of antibody levels.

Seven studies included healthy controls (HC) in the study cohorts, while two studies [4,28] did not employ controls. All studies accounted for the confounding factors of age, anticancer treatment, and steroid use, while most also included performance status and comorbidities [4,30,31,32,34,35]. Overall, after primary dose vaccination, patients with lung cancer showed lower seropositivity rates compared to healthy controls. In some cases, suboptimal response to the vaccine was observed in a certain percentage of lung cancer patients. It has also been shown that additional booster doses of the COVID-19 vaccination can improve the inferior immunological effectiveness of the primary doses in patients with lung cancer. Five studies reported seroconversion rates (defined as the proportion of vaccine recipients whose blood titers for SARS-CoV-2 vaccine antibodies were higher than the test-specific lower limit cutoff value) of LC or thoracic cancer patients at 93.7% [29], 95.2% [4,30], 98% [30], 99.1% [28], and 96.7% [31,32]. Mack et al. [32] reported that LC patient Ab titers had a significantly reduced area under the curve per day compared to control antibody titers. Two studies, Nakashima et al. [30] and Hibino et al. [31] reported that there was no difference in the percentage of seroconversion between LC patients and HC. 

One of the first studies to focus on thoracic malignancies was by Gounant et al. [29]. In this study, conducted in France, 306 LC patients were followed after receiving the first and second doses of the vaccine. At 28 days after receiving the first dose, 32.3% of the patients still displayed no anti-S IgG levels. An overall rise in serum anti-S IgG titers was observed between 2 and 9 weeks after administration of the second dose. However, the median serum anti-S was still significantly lower than controls, and 6.3% of patients still exhibited negative anti-S titers. A small percentage of patients who had low antibody titers after the second dose of the vaccine were given a third dose; 88.5% of them showed seroconversion. Some of the 11.5% who did not respond to the third dose had concomitant hematologic conditions (hypogammaglobulinemia, monoclonal IgG peak), which might explain their lack of response to immunization. 

In a longitudinal study, Valanparambil et al. [33] examined the serological response to COVID-19 vaccination in patients with non-small cell lung cancer (NSCLC). In a cohort with 82 patients, the majority produced a response comparable to the group of healthy controls one month after mRNA vaccination. However, a small subset produced poor seroconversion, with 25% having markedly lower binding antibody titers. The study found a significant decline in the anti-spike titers approximately 6 months after the second dose. The LC patients who received a third booster shot showed significantly increased binding antibody titers around 5–30 days after, although this study also describes a decrease in binding IgG titers 60–110 days subsequent to the third booster. 

Similarly, Mack et al. [32] investigated the serological response by assessing immunogenicity every 3 months for a year. In the cohort of 114 LC patients, 66% had adenocarcinoma, and 68% were receiving systemic treatment. Although a large majority of LC patients mounted a similar antibody response compared to healthy participants, LC patients had a significantly reduced area under the curve per day compared to the control group, *p =* 0.0018. This suggests that while most LC patients did produce antibodies in response to the vaccine, their antibody maintenance over time was lower than HC. Interestingly, 5% of post-vaccination LC patients had titers below the level of detection, in contrast to zero percent in the control population. This study serves as further evidence of the value of the third vaccination dose, as all of the patients who received it experienced a considerable rise in antibody titers. 

Another similar study conducted by Hernandez et al. [28] followed 126 LC patients with time points before and every 3 months after the vaccination to measure IgG antibodies. At 3 to 6 months after the vaccination, 99.1% of LC patients had seroconverted, which was the highest seroconversion rate among the reviewed studies. Most of the patients developed immunity after the first and second doses. IgG titers were maintained over time, with low infection and reinfection rates with a mild clinical course. None of the reports by Valanparambil et al. [33], Mack et al. [32], and Hernandez et al. [28] observed differences in serologic responses as a result of cancer treatment in LC patients, however, the significant heterogeneity in treatment modalities may indicate they were not necessarily powered to do so.

Bowes et al. evaluated the immunogenicity of SARS-CoV-2 vaccines in a prospective cohort of 33 lung cancer patients receiving radiotherapy compared to two different control groups of 181 LC patients with no radiotherapy and 187 healthy controls [34]. According to this study, LC patients receiving radiotherapy showed lower anti-SARS-CoV-2 spike antibody titers than the other two groups. However, a more detailed analysis revealed that the concurrence of immunosuppressive conditions (involving the use of immunosuppressive drugs concurrently or the presence of immunosuppressive diseases such as chronic lymphocytic leukemia) in these patients, could be, in turn, contributing to this low serological response, suggesting that radiotherapy itself may not be impacting on antibody levels. 

Trontzas et al. [35] described the longitudinal humoral response in the LC cohort of 125 patients and compared their magnitude to a group of solid cancer patients and healthcare workers (HCWs). For LC patients, antibody response peaked around 6 weeks after the second dose of the COVID-19 vaccine and declined thereafter, being significantly lower compared to the group of healthy participants. In this study, they also showed that certain clinicopathological features, such as active smoking, were related to significantly lower antibody titers in LC patients. Hibino et al. [31] investigated 126 LC patients and 65 control patients who visited the outpatient department of respiratory medicine and rheumatology. Interestingly, all LC patients received immune checkpoint inhibitors (ICIs), with the plurality (49.2%) receiving pembrolizumab. Among the 65 LC patients analyzed, 96.7% showed seroconversion compared to 100% of the 65 healthy controls showing seropositivity, not significantly different. However, the median values of anti-spike antibodies were significantly lower in patients with lung cancer: 3238.6 AU/mL versus control patients, 12,260.7 AU/mL (*p* < 0.001). 

One of the largest studies to date, a non-peer-reviewed preprint, is a prospective, longitudinal study conducted in 37 hospitals in Spain. Provencio et al. [4] evaluated the immunogenicity of the COVID-19 vaccination in 1976 LC patients, where 71.4% had adenocarcinoma and 82.6% had active treatment. Two weeks after completion of the primary vaccination doses, 95.2% of LC patients responded to the vaccine with a geometric mean titer of 655.45 BAU/mL (seropositivity SARS-CoV-2 S-RBD IgG cutoff > 7.1 BAU/mL), leaving 4.8% of LC patients with a presumed compromised serological response. Patients with performance status 2 or higher and/or comorbidities had a greater likelihood of not mounting an antibody response after vaccination. Significant differences were observed depending on the type of therapy received. Specifically, patients treated with immunotherapy or oral targeted therapy had a lower probability of being seronegative than those treated with chemotherapy (OR 0.26; *p* < 0.001 and OR 0.13; *p* < 0.001, respectively). The high-dose systemic steroids often given to cancer patients under chemotherapy may partially explain this observation. However, there were no significant differences between patients receiving active therapy and those not.

In another study, Nakashima et al. [30] investigated the immunogenicity of the BNT162b2 mRNA vaccine in a cohort of 55 LC patients, with a mean age of 73.1, the oldest mean age among all our eligible studies. This was one of two studies that showed no difference in seropositivity rate between LC patients and healthy controls, at 98% and 100%, respectively. However, LC patients had significantly lower seroprotection rates, as only 64% of LC patients mounted an equal to or greater than the cutoff point of 1084 AU/mL compared to healthy controls (87% seroprotection rate) (*p =* 0.017) using the Architect SARS-CoV-2 IgG II Quant assay from Abbott Laboratories. The Elecsys Anti-SARS-CoV-2 assay (Roche Diagnostics) obtained the same results, with only 66% of LC patients showing seroprotection compared to healthy controls (90%) (*p =* 0.013). In their analysis of the anticancer treatment types, lung cancer patients receiving cytotoxic agents had a significantly lower adjusted odds ratio for both seropositivity and seroprotection (>1084 AU/mL for Architect and >150 AU/mL for Elecsys) compared to non-cancer patients.

It is important to note that no correlation to protection has yet been identified to determine the clinical effectiveness of immune responses. Several studies have suggested that the presence of neutralizing antibodies (NT Ab) could be a reliable biomarker for predicting resistance to SARS-CoV-2 infection since it is associated with rapid clearance of the virus [36,37,38]. However, the lack of harmonization in the calibration of the assays makes it difficult to identify the precise concentration of neutralizing antibodies required for protection, with assay variations rendering the comparison across studies challenging. 

Out of the nine identified papers, three reported neutralizing antibodies as an endpoint. Bowes et al. studied neutralization activity after two doses of SARS-CoV-2 vaccines in 20 of 33 lung cancer patients receiving radiotherapy [34]. This study showed that 25% of the patients analyzed for neutralization activity had a titer level lower than the threshold associated with 50% protection (20% of the GMT geometric mean neutralizing titer [37]. In order to test whether healthy vaccine recipients and NSCLC patients had neutralizing antibodies against the parental virus and a VOC in response to the first doses of mRNA vaccines, Valanparambil et al. [33] used a live virus neutralization assay. They found that 25% of patients had lower NT Ab titers than controls against WT and that a significant fraction of LC patients (18%) failed to generate detectable NT Ab titers. Moreover, the authors showed that the capacity to neutralize the Omicron variant was compromised. An increase in NT Ab titers was observed after the third dose, but this declined after 60–110 days after the booster shot. In a subset of 28 LC patients, Mack et al. [32] also aimed to assess whether the third mRNA vaccine-induced immune response protects against SARS-CoV-2 variants. This study demonstrated that both patients and healthy controls had considerably lower Omicron neutralization abilities compared to the wild-type, and 21% of LC patients had no detectable NT Ab against Omicron. 

The cellular response as a biological variable associated with immunization was measured in only one paper by Gounant et al. [29] in a small subset of patients. At day 28 after vaccination, both T-lymphocyte CD3+ and CD4+ counts were associated with seroconversion when using a cutoff of 50 AU/mL (*p* < 0.01, and *p* = 0.01, respectively). However, this association did not extend to the 300 AU/mL seroconversion cutoff. The association with higher seroconversion probability was observed by measuring interferon-γ-specific T-cell response to the SARS-CoV-2 spike using both cutoffs. 

The studies reviewed consistently report lower seropositivity rates in LC patients compared to HC after receiving primary doses of the SARS-CoV-2 vaccine. Although certain variables, such as age, long-term corticosteroid use, active smoking, performance status > 2, comorbidities, and cytotoxic agent use, have been associated with poor immunization [4,29,30,33,35], the underlying biological mechanisms are not fully understood. The authors encourage additional longitudinal research with larger cohorts in order to pinpoint the factors that are responsible for this effect. These studies should also evaluate vaccine-induced T-cell responses in patients to determine if both cellular and humoral immune responses are compromised. A better understanding of these issues will help optimize vaccination strategies for patients with lung cancer. 

As SARS-CoV-2 continues to evolve, resulting in the emergence of multiple variants of concern and variants of interest, researchers and healthcare professionals have developed new vaccination strategies to combat these new variants. It is essential to continue studying immune responses to emerging variants so that appropriate countermeasures can be enabled swiftly and efficiently. It is also crucial to highlight that the studies presented in these publications were carried out prior to the authorization or availability of the bivalent COVID-19 vaccine. 

## 3. SARS-CoV-2 Vaccines Immunogenicity in Patients with Cancer

The literature also has a large number of research investigations into the immunogenicity of the SARS-CoV-2 immunization in solid cancer patients, with a strong representation of patients with lung cancer. Studies evaluating the serologic response in patients with solid tumors and/or hematologic cancer have been included in the review where the percentage of patients with lung cancer was equal to or greater than 10% and included specific data on immunogenicity in these patients. Among these studies, LC patients were a mean of 18 ± 9.8% of the population. 

Lasagna et al. [39] (58/142, 41% LC) evaluated the immunogenicity of the BNT162b2 third dose in a cohort of 142 solid cancer patients with 41% LC patients. Measurements were taken before and three weeks after getting the third dose. Before the third dose, 16.2% of the subjects showed negative antibody levels. After the booster shot, the median levels went from 157 BAU/mL to 2080 BAU/mL, and all except one patient showed detectable IgG ab levels. This study also assessed neutralizing antibody levels. Before the third dose, 18.3% of LC patients were negative for NT Ab. After the booster, a 16-fold increase was observed. However, there was a 32-fold lower NT Ab against Omicron compared to the wild-type strain (*p =* 0.0004) and a 12-fold lower compared to the Delta strain (*p =* 0.0110). This study also evaluated the cellular response through the SARS-CoV-2 Interferon Gamma Release Assay in 72 paired samples, finding that 41/72 were positive for IFN-g after receiving the third dose.

A study conducted by Linardou et al. [40], comprising 189 solid cancer patients (57/189 30% LC) and 189 healthy controls, also looked at seropositivity rates and factors that could influence vaccine response, using BNT162b2 (86.2%), mRNA-1273 (10.1%), and AZD1222 (3.7%). They measured titers 2–4 weeks after the second vaccine was administered. Using a seropositivity cutoff of 38.8 BAU/mL, they found 90.5% of cancer patients to be seropositive, while 98% of healthy controls were seropositive. When using a cutoff of 488.8 BAU/mL for seroprotection, the difference was highlighted even more, with 54.5% of cancer patients considered seroprotected versus 97% in healthy controls. They found multiple clinical factors that impacted immunogenicity. Older age, poorer performance status, active treatment, smoking status, and SCLC or pancreatic cancer were all associated with reduced titers. 

Mencoboni et al. [41] (22/195 11%LC) investigated the serological response to COVID-19 mRNA vaccines (Pfizer and Moderna administered to 56% and 44% of subjects, respectively) in 195 cancer patients receiving chemotherapy compared to 400 subjects randomly selected from the HCWs group (control). Amongst the solid tumor groups, those with lung cancer (*N* = 22) showed the poorest serological response with a mean concentration of 257.7 BAU/mL, in contrast to 319.3 BAU/mL for the control group. In a study looking at 816 solid cancer patients, of which 168 (21%) were lung cancer patients, compared to 274 healthy controls, Di Noia et al. [42] also found a reduced response to the first and second BNT162b2 vaccine doses. Levels greater than 15 AU/mL were set as the cutoff for a response, and >80 AU/mL defined a strong response. At seven weeks after the first vaccination, 94.2% of cancer patients were found to respond, and 86% were found to have a strong response, while 100% of healthy controls responded, with 99.2% exhibiting a strong response. However, LC patients were not examined independently. The same authors presented the results after six months of follow-up [43], showing a rapid decline in humoral response in their cohort of 400 solid cancer patients (83/400 20.8% LC) versus 232 HCWs. In a third report [44], the study team described that after receiving the third dose of the BNT162b2 vaccine, the vast majority of patients (98.8%) had positive serological readings. However, five patients did not respond to either the second or third dose, and further investigation will be needed to assess their cellular immunity. According to their analyses, only chronic steroid use was significantly linked with reduced antibody levels. However, there was no noticeable difference in the titer depending on the anticancer treatment type.

Zeng et al. [45] (29/160 18% LC) found that after two doses of mRNA vaccine, patients with lung cancer had lower neutralizing antibody responses when compared to HCWs. Specifically, 61% of cancer patients and 52% of the HCWs received BNT162b2, and 39% and 48% received mRNA-1273, respectively. Overall, cancer patients exhibited a 2.4-fold lower neutralizing antibody response than HCWs. The mean neutralizing antibody titers for LC patients was 307, compared to 522 for HCWs. Notably, 31% of LC patients showed no detectable neutralizing antibody response (NT50 values below 40). 

Quantification of spike-binding antibody titers in numerous studies has also shown reduced median IgG titers in patients with solid cancer compared to healthy volunteers. For example, Massarweh et al. [46] (26/102, 25% LC) evaluated the seropositivity rates in 102 adult patients with solid tumors, of which 25% were LC patients. The binding antibody response rate following two doses of the BNT162b2 vaccine was significantly lower compared with the controls, at 90% and 100%, respectively. Among the 9.8% who were seronegative, there were two LC patients. In addition, the median IgG titer for cancer patients was statistically significantly lower than the control group. For LC, it was lower than the median for all solid tumor patients.

Amatu et al. [47] examined seropositivity in 171 solid tumor cancer patients (27/171 16% LC) and 2406 healthy controls. After two doses of SARS-CoV-2 mRNA vaccines, 94.2% of cancer patients were seropositive using a cutoff of 33.8 BAU/mL. BNT162b2 was administered to 88% of the patients, while mRNA-1273 was administered to 12%. In comparison, 99.8% of healthy controls were seropositive, all receiving BNT162b2. Looking specifically at LC patients, 24/27, or 89%, were seropositive. However, in contrast to the study by Linardou et al. [40], this study team found that ECOG PS > 2 was the only variable affecting seropositivity and found no correlation with age, steroid therapy, BMI, gender, cancer staging, or anticancer treatment. Of note, none of the patients received over 25mg/day of prednisone. 

Webber et al. [48] (30/291, 10% LC) aimed to find predictors for poor seroconversion to the SARS-CoV-2 mRNA BNT162b2 vaccine in patients with solid tumors. They concluded a strong association of poor seroconversion with current treatment and types of therapies, with those receiving chemotherapy, hormone therapy, and targeted therapy showing the highest rates at 13.9%, 11.4%, and 21.7%, respectively. Amongst the patients in the study (*N* = 291), 30 were lung cancer patients, of which 4 (13.3%) showed poor serological responses. 

Tal Goshen-Lago et al. [49] (45/232 19% LC) analyzed the short-term seroconversion rate following two doses of the BNT162b2 vaccine in a cohort of 232 cancer patients. They described that the seropositive rate reached 86% (*N* = 187) in the patients. Among LC patients, 6/43 remained seronegative. The same study team of Waldhorn et al. [50] (36/154, 23% LC) also aimed to evaluate the mRNA-based BNT162b2 vaccine immunogenicity at six months post-vaccination in 154 patients (23% LC) and 135 controls. Similar to Di Noia et al. [42], this study showed titer decay in both cohorts. 

In a similar study, Ligumsky et al. [51] (45/326 14% LC) analyzed 326 patients treated with anticancer therapy to assess the immunogenicity to SARS-CoV-2 following two doses of the BNT162b2 vaccine compared to a group of 164 healthy adults. According to the study, 11.9% of cancer patients were seronegative; within this subset, 12.8 % were LC patients. Moreover, comparing cancer patients to controls, median IgG titers were statistically substantially lower in the group with cancer. Furthermore, this study showed a relationship between antibody levels and therapy, indicating that the seronegative proportions were higher in the chemotherapy-treated group than in those receiving ICI or targeted therapy.

In line with this study, Grinshpun et al. [52] (38/202 19% LC) measured antibodies against the SARS-CoV-2 mRNA vaccine (BNT162b2) in 202 cancer patients on active treatment. In comparison to 100% in the control group, their study revealed that the seroconversion rate following vaccination was 89.1% in patients with a history of COVID-19 and 87.2% in patients without a history of COVID-19 (*p* < 0.001). Similar to the previous report, this research revealed that only the chemotherapy regimen was independently related to significantly diminished humoral response to infection or immunization in univariate analysis.

The study conducted by Figueiredo et al. [53] (13/291, 4.5% LC) investigated longitudinal changes in seropositivity and antibody levels in 291 cancer patients receiving two doses of either BNT162b2 or mRNA-1273 vaccines. It showed that patients with thoracic cancer had the lowest percentage of high seropositivity maintenance (equal to or greater than 4160 AU/mL) among those with solid cancer at a median follow-up of 42 days after the second dose. 

## 4. Summary

The clinical development of the SARS-CoV-2 vaccine advanced with unprecedented speed. However, cancer patients were not included in the pivotal clinical trials, requiring additional focused studies on select cancer patient populations. As a first approach, observational studies concentrated their efforts on the assessment of the antibody response. Articles that focused only on LC patients showed that COVID-19 mRNA and viral vector-based vaccines could safely generate humoral immune responses in these patients, albeit at somewhat diminished levels compared to the general population. Out of the nine articles that specifically examine vaccine immunogenicity in LC patients, six demonstrated significant differences in median titers following the primary vaccination series when compared to HC data [29,30,31,33,34,35]. The variety of emerging and still-evolving Omicron variants of SARS-CoV-2, with the ability to overcome primary vaccination-induced immunity, present significant new threats to LC patients. On a positive note, reports have shown that a third mRNA vaccine dose can improve the lower immunological response of the primary doses in LC patients. Despite this, Valanparambil et al. [33] and Mack et al. [32] noted that a fraction of LC patients had suboptimal neutralizing antibody titers against Omicron than to the wild-type strain. The articles focused on solid cancer followed the same trend. Of the 12 research teams, 6 found that after the second dose of the vaccine, seroconversion levels were lower in cancer patients than in HC patients [40,42,46,47,49,52]. The Massarweh and Ligumsky publications [46,51] further noted that the mean IgG titers were significantly lower in the cancer group. Mencoboni, Massarweh, and Ligumsky pointed out that those with LC showed the poorest serological response amongst solid cancer patients [41,46,51]. Linardou et al. found that SCLC was associated with reduced titers [40]. Figueiredo revealed that only a small number of patients with thoracic cancer retained strong seropositivity. Zeng et al. further showed that LC patients had the lowest neutralizing antibody response compared to HC after two vaccine doses [53]. Six months after receiving the primary doses, two groups observed titer decay [43,50]. After the third dose, Di Noia et al. described that most patients had positive serological readings [44], and Lasagna et al. [39] observed a 16-fold increase in neutralizing antibody levels. However, these were significantly lower against Omicron compared to the wild-type strain. Among these investigations, some found that clinical factors, such as older age, poorer performance status, active treatment, smoking [40], and ECOG PS > 2 [47], were influencing immunogenicity in cancer patients. Webber et al. found a strong association with low seroconversion in the groups undergoing chemotherapy, hormone therapy, and targeted therapy [48]. Additionally, according to research by Ligumsky and A. Grinshpun, chemotherapy was associated with a lowered humoral response [51,52].

## 5. Conclusions

The COVID-19 pandemic has brought uncertainty and fear to many, especially those with pre-existing conditions such as lung cancer. The SARS-CoV-2 vaccines have been proven to be both safe and effective in clinical trials, and research has shown that the majority of LC patients mount a comparable response to healthy individuals. Furthermore, it has not been observed to interfere with the therapy of lung cancer patients. It is therefore advised that lung cancer patients, who are classed as immunocompromised by the CDC [54], follow the suggested vaccination schedule.

There is currently limited evidence linking the deficient vaccine immune response observed in LC patients to any specific clinicopathological feature or particular course of therapy. The importance of this topic calls for continued research, such as prospective studies looking at large populations with diverse treatment settings, particularly in response to the bivalent vaccine. 

Research has shown that SARS-CoV-2 vaccines can stimulate antibody production and activate cellular immunity. However, it remains unclear how this response may differ in cancer patients. Until now, research has primarily focused on humoral immune reactions in LC patients. Ongoing research on lung cancer patients’ cellular immune responses to vaccinations will be able to provide insight into whether or not these patients have an optimal response that helps to fight COVID-19 infection. 

Longitudinal prospective research is needed to fill in the significant gaps in the literature on what happens to those subsets with suboptimal immune responses, the chance of breakthrough infections for these patients, and how cellular immunity develops. Identifying these elements will help improve COVID-19 vaccination strategies for the most vulnerable populations.

## Figures and Tables

**Table 1 vaccines-11-00969-t001:** Summary of studies evaluating SARS-CoV-2 vaccine responses in patients with lung cancer.

Year, Citation, Reference	Vaccination Types (%) for LC Pt.	Test Groups (Cohort, *N*) ^a^	Histological Classification (Type, *N* (% )) ^c^	Age (Median) ^d^	Treatment Stratification (Treatment, *N* (%)) ^e^	Time Points	Serological Assay (Cutoff, Manufacturer)	Seroconversion Rate (%) or Median Binding Antibody Titers ^g^	Compromised Titer Pt. ^h^
2022 Provencio et al. [4]	BNT162b2 (39.6%) mRNA-1273 (54.1%) AZD1222 (5.2%) Ad26. COV2.S (0.4%) Other (0.6%)	LC Pt., *N* = 1976	LUAD, *N* = 1080 (71.4%) SCC, *N* = 335 (22.1%) ASC, *N* = 14 (0.9%) NOS, *N* = 53 (3.5%) Other, *N* = 31 (2%)	LC Pt., mean: 66.7	CTx, *N* = 531 (38.2%) IO, *N* = 605 (43.5%) Oral TT, *N* = 228 (16.4%) Other, *N* = 26 (1.9%)	2 weeks after second dose	SARS-CoV-2 S-RBD IgG CLIA (>7.1 BAU/mL, Snibe)	LC Pt., seroconversion rate: 95.2% LC Pt., GMC of 655.45 BAU/mL	4.8% seronegative Pts.
2022 Hernandez et al. [28]	mRNA-1273 (94.4%) mRNA-1273 and BNT1612b (1.6%)	LC Pt., *N* = 126	NSCLC, *N* = 111 (88.1%) LCNEC, *N* = 2 (1.6%) Carcinoid, *N* = 1 (0.8%) SCLC, *N* = 12 (9.5%)	LC Pt., 66	CTx, 23% CTx and IO, 14% IO, 35% TKI, 20% Surveillance/ Other treatment, 8%	3–6 months after second dose	COVID-19 quantitative IgG ELISA (>40 UI/mL, Demeditec Diagnostics)	LC Pt., seroconversion: 99.1% LC Pt., median titer: 720 IU/mL	
2021 Gounant et al. [29]	BNT162b2 (98%) mRNA-1273 (0.3%) AZD1222 (1%)	Thorac. cancer Pt., *N* = 306 HC, *N* = 18	NSCLC, *N* = 260 (84.9%) SCLC, *N* = 22 (7.2%) MPM, *N* = 13 (4.2%) Others, *N* = 11 (3.5%)	Thorac. cancer Pt., 67	CTx, *N* = 74 (24.2%) ICI, *N* = 70 (22.9%) Oral TKI/Bevacizumab, *N* = 43 (13.7%) RT, *N* = 2 (0.6%) No treatment, *N* = 141 (46.1%)	Median days: 52 after second dose	Architect SARS-CoV-2 IgG II Quant (≥50 AU/mL, Abbott)	Thorac. cancer Pt., seroconversion: 93.7% Thorac. cancer Pt., median titer: 4725 AU/mL * HC, median titer: 10,594 AU/mL*	6.3% seronegative Pts. 11% Pts. ≤300 AU/mL
2022 Nakashima et al. [30]	BNT162b2 (100%)	LC Pt., *N* = 55 HC, *N* = 38	LUAD, *N* = 44 (83%) SCC, *N* = 4 (8%) SCLC, *N* = 5 (9%)	LC Pt., mean: 73.1 HC, mean: 71.2	Cytotoxic agent, *N* = 13 (15%) TKI, *N* = 20 (38%) Cytotoxic agent and ICI, *N* = 8 (15%) ICI, *N* = 11 (21%) Angiogenesis inhibitor, *N* = 7 (11%)	3–5 weeks after second dose	Architect SARS-CoV-2 IgG II Quant (≥ 50 AU/mL, Abbott) Elecsys Anti-SARS-CoV-2 S (≥0.8 U/mL, Roche)	LC Pt., seroconversion rate: 98% HC, seroconversion rate: 100% LC Pt., GMC: 1632 AU/mL * HC, GMC: 3472 AU/mL *	
2022 Hibino et al. [31]	BNT162b2 (98.4%) mRNA-1273 (1.6%)	LC Pt., *N* = 126 Control Pt., *N =* 65 ^b^	LUAD, *N* = 73 (57.9%) SCC, *N* = 38 (30.2%) SCLC, *N* = 4 (3.2%) Others, *N* = 11 (8.7%)	LC Pt., 71	Pembrolizumab, *N* = 62 (49.2%) Nivolumab, *N* = 10 (7.9%) Atezolizumab, *N* = 28 (22.2%) Durvalumab, *N* = 21 (16.7%) Nivolumab and Ipilimumab, *N* = 5 (4%)	Median days: 14 after second dose	Architect SARS-CoV-2 IgG II Quant (≥50 AU/mL, Abbott)	LC Pt. with ICI, seroconversion: 96.7% Control Pt., seroconversion: 100% LC Pt., median titer: 3238.6 AU/mL* Control Pt., median titer: 12,260.7 AU/mL *	
2022 Mack et al. [32]	BNT162b2 (75%) mRNA-1273 (34%)	LC Pt., *N* = 114 HC, *N* = 114	LUAD, *N* = 75 (66%) SCC, *N* = 17 (15%), NSCLC NOS, *N* = 9 (8%) SCLC, *N* = 10 (9%) Other, *N* = 2 (1%) Mixed, *N* = 1 (1%)	LC Pt., 69 HC, 62	CTx, *N* = 17 (15%) IO, *N* = 18 (16%) CTx and IO, *N* = 24 (21%) TT, *N* = 16 (14%) TT and CTx, *N* = 3 (3%) No systemic therapy, *N* = 36 (32%)	Every 3 months independent of vaccination date	In-house two-step ELISA (OD490 nm > 0.15) ^f^	LC Pts. Ab titers had a significantly reduced area under the curve per day compared to HC *	5% LC Pts. with 0 titers measurements
2022 Valanparambil et al. [33]	BNT162b2 (68%) mRNA-1273 (32%)	LC Pt., *N* = 82 HC, *N* = 53	NSCLC, *N* = 82 (100%)	LC Pt., 68 HC, 34	CTx, *N* = 32 (48%) IO, *N* = 38 (56%) TT, *N* = 23 (34%) RT, *N* = 3 (4.8%)	1 month after second dose	V-PLEX COVID-19 Respiratory Panel 2 Kit (1000 Relative Luminescence Unit, MSD)	LC Pt., median titer: 4.94 Log10 AU/mL * HC, median titer: 5.46 Log10 AU/mL *	25% lower titer Pts.
2021 Bowes et al. [34]	BNT162b2 (64%) mRNA-1273 (30%) Ad26. COV2.S (1%)	LC Pt. with RT, *N* = 33 LC Pt. without RT, *N* = 181 HC, *N* = 187	NSCLC, *N* = 31 (94%) SCLC, *N* = 2 (6%)	Pt. with RT, 68	RT, *N* = 33 No RT, *N* = 181	Median days: 87 after second dose	Elecsys Anti-SARS-CoV-2 S (≥0.8 U/mL, Roche)	LC Pt. with RT, GMC of 2.42 Log10 U/mL * LC Pt. no-RT, GMC of 2.62 Log10 U/mL HC, GMC: 2.80 Log10 U/mL *	25% Pts. with titer <20% of GMT
2022 Trontzas et al. [35]	BNT162b2 (80%) mRNA-1273 (8%) AZD1222 (12%)	LC Pt., *N* = 125 Solid tumor Pt., *N* = 35 HC, *N* = 86	NSCLC, *N* = 104 (83.2%) SCLC, *N* = 21 (16.8%)	LC Pt., 68 Solid tumor Pt., 59 HC, 50	IO, *N* = 60 (48%), CTx, *N* = 34 (27.2%) IO-CTx, *N* = 29 (23.2%) TT (TKI), *N* = 2 (1.6%)	21–27 weeks after first dose	Elecsys Anti-SARS-CoV-2 S (≥0.8 IU/mL, Roche)	LC Pt., median titer: 141 IU/mL *	Active smokers LC Pt. showed reduced Ab

^a^ HC: healthy controls; RT: radiotherapy. ^b^ Patients who visited outpatient departments of respiratory medicine and rheumatology. ^c^ NSCLC: non-small cell lung cancer; SCLC: small cell lung cancer; MPM: malignant pleural mesothelioma; ASC: adenosquamous carcinoma; LUAD: lung adenocarcinoma; NOS: not otherwise specified/undifferentiated; SCC: squamous carcinoma; LCNEC: large cell neuroendocrine carcinoma. ^d^ Median age values are presented unless otherwise stated. ^e^ CTx: Chemotherapy; IO: Immunotherapy; ICI: Immune checkpoint inhibitor; TKI: Tyrosine Kinase Inhibitor Therapy; TT: Targeted therapy. ^f^ ELISA: enzyme-linked immunosorbent assay, cut-off: optical density at 490 nm (OD490) > 0.15 at 1:80 plasma dilution were considered positive. ^g^ GMC: Geometric mean spike antibody concentration; *: statistical differences were found compared to HC data. ^h^ GMT: geometric mean neutralization titer.

## Data Availability

No new data were created.

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
