# Peer review of "Serological Response to SARS-CoV-2 after COVID-19 Vaccination in Lung Cancer Patients: Short Review"

_vaccines, 2023, doi:10.3390/vaccines11050969_

Round 1

Reviewer 1 Report

Authors have described findings of humoral responses to covid-19 vaccines for lung cancers participants. They used a narrative approach as available studies differed in follow up time and cut-off points for immune response. Findings indicated immune responses that compares to controls, though with fast decline of immune responses, highlights the need for future large prospective studies.

Overall the manuscript is well written and organized. Authors can revise contents of Table 1 and add the references which are missing throughout the first column

Author Response

 Thank you for your review and comments. We revised Table 1 and included missing references in the first column.   

Reviewer 2 Report

The authors have made an interesting attempt at “Serological Response to SARS-CoV-2 after COVID-19 Vaccination in Lung Cancer patients: Short Review.” The manuscript is interesting; however, the authors need to justify the scientific writing manuscript. Some of the general comments are provided below:

1.     What were the specific mRNA and viral vector-based vaccines that were investigated in the studies? Did the studies assess the immunogenicity of different types of vaccines or only specific ones?

2.     What could be the potential reasons for the lower seropositivity rates observed in lung cancer patients compared to healthy controls after primary dose COVID-19 vaccination? Are there differences in the immune response or vaccine effectiveness in lung cancer patients that may contribute to this observation?

3.     What are the implications of the reduced area under the curve per day of antibody titers in lung cancer patients compared to healthy controls, as reported by Mack et al. [32]? What could be the potential clinical significance of this finding in terms of vaccine effectiveness and protection against SARS-CoV-2 infection?

4.     What were the characteristics of the cancer patients included in the studies, such as age, performance status, and types of cancer treatments received? Did the studies account for potential confounding factors, such as comorbidities or concurrent medications, that may have influenced the immunogenicity of the vaccines in cancer patients?

5.     Did the studies investigate the impact of the Omicron variant or other emerging variants of SARS-CoV-2 on the immunogenicity of the vaccines in cancer patients, including those with lung cancer? Were there any differences in the antibody responses against different variants of the virus?

6.     Did the studies investigate other immune responses beyond antibodies, such as cellular immune responses or T cell responses, in cancer patients after vaccination?

Author Response

We appreciate your thorough comments and please find the attached letter and point-to-point response.  

Reviewer 3 Report

The manuscript is clear and well written.

Author Response

Thank you for your review and comments. 

Round 2

Reviewer 2 Report

The authors have modified the manuscript and now it is acceptable for publication.